# Respiratory Symptoms, Allergies, and Environmental Exposures in Children with and without Asthma

**DOI:** 10.3390/ijerph191811180

**Published:** 2022-09-06

**Authors:** Agata Wypych-Ślusarska, Martina Grot, Maria Kujawińska, Maciej Nigowski, Karolina Krupa-Kotara, Klaudia Oleksiuk, Joanna Głogowska-Ligus, Mateusz Grajek

**Affiliations:** 1Department of Epidemiology, Faculty of Health Sciences in Bytom, Medical University of Silesia in Katowice, 40-055 Katowice, Poland; 2Department of Public Health, Department of Public Health Policy, Faculty of Health Sciences in Bytom, Medical University of Silesia in Katowice, 40-055 Katowice, Poland

**Keywords:** bronchial asthma, allergy, environmental factors, children, symptoms

## Abstract

Background: Epidemiological data concerning the level of asthma morbidity indicate that in Poland, asthma is diagnosed in 5–10% of the pediatric population. Aim The purpose of this study was to compare the prevalence of respiratory symptoms and allergies in a group of children with and without asthma and to evaluate the association between exposure to environmental factors and the prevalence of bronchial asthma in a pediatric population. Material and Methods: A cross-sectional study was conducted on a group of 995 children attending primary schools in the province of Silesia in 2018–2019. The research tool was an anonymous questionnaire developed based on the form used in The International Study of Asthma and Allergies in Childhood (ISAAC). Children’s health status, the prevalence of bronchial asthma, and the performance of allergic skin tests were assessed based on parents’ indications in a questionnaire. Environmental exposures such as mold and dampness in apartments or ETS were similarly assessed. Analyses were performed using Statistica 13.0; *p* < 0.05. Results: A total of 88 subjects (8.8%) suffered from bronchial asthma. Parents of children with asthma, compared to parents of children without the disease, were more likely to rate their children’s health as rather good (43.2% vs. 38.0%) or average (21.6% vs. 3.1%). All analyzed respiratory symptoms, as well as allergies, were statistically more frequent in children with bronchial asthma. Conclusions: The parent’s subjective assessment of the child’s health varied significantly according to the asthma diagnosis. Asthma is also associated with other diseases: allergic reactions to pollen, house dust, hay fever, and AD (atopic dermatitis) were statistically significantly more frequent among children diagnosed with bronchial asthma.

## 1. Introduction

According to *The Global Initiative for Asthma* 2021 (GINA), asthma is defined as a heterogeneous disease characterized by chronic inflammation of the respiratory system. Epidemiological data on the prevalence of asthma in Poland indicate that it is diagnosed in 5–10% of the pediatric population. The nationwide cohort study ECAP (*Epidemiology of allergic diseases in Poland*) confirms the incidence of this disease in the pediatric population aged 6–7 years at the level of 4.4% among children living in urban areas and 3.9% living in rural areas [1,2].

The clinical classification of asthma can be made according to different types of parameters. The most common phenotypes of asthma are allergic asthma, non-allergic asthma, adult-onset asthma, asthma with persistent airflow limitation, and asthma with obesity [1]. In allergic asthma, the onset of the disease usually occurs in childhood, and comorbidities such as AD (atopic dermatitis), allergic conjunctivitis, eczema, allergic rhinitis, and gastrointestinal allergy are also observed, whereas, in non-allergic asthma, the diagnosis is usually made in adults. The clinical picture of pediatric patients with a diagnosis of asthma is on physical examination with the following symptoms: wheezing, perceived dyspnea, chest tightness, and a reflex response from the airways accompanied by a characteristic sound as a result of an antagonistic mechanism towards sudden glottal closure (cough) [1,2,3,4]. The most common symptoms associated with asthma are wheezing, dyspnea, and cough. However, a single symptom is not a sufficient predictor of asthma, as wheezing also occurs in children without the disease. Environmental risk factors are one of the biggest determinants of asthma prevalence [5,6]. Epidemiological studies confirm that one of the major risk factors for asthma is environmental tobacco smoke (ETS). There is also growing evidence that molds are a risk factor, although some studies suggest that molds are only harmful in the bedroom but have a protective effect elsewhere in the house [7]. In contrast, there is controversy about the role of animal allergens in asthma. Some studies suggest a protective role for animal allergens [7], while others suggest that they are a risk factor for the development of the disease [8].

Asthma, as well as allergic reactions and hypersensitivity states, are an area of scientific research showing a correlation between pulmonology and allergology as a result of, among others, epigenetic predictors during DNA strand methylation and high rates of IgE-dependent antibodies [9,10]. Asthma and allergic diseases affect boys more often than girls. The reasons for these differences may be due to the different structures of the female and male lungs. Female lungs contain a greater number of innate type 2 lymphoid cells, which are part of natural immunity. Also important are differences in hormonal balance, which may activate inflammatory mechanisms associated with macrophage polarization [11,12].

The etiopathogenesis of asthma and allergic reactions is explained by the hygiene hypothesis, the assumptions of which are based on the increase in the degree of morbidity through the implementation of excessive personal hygiene rules, sanitary regime, and increased degree of agglomeration and unbalanced quantitative-qualitative diet. The latter factor can be categorized as environmental conditions. Diet affects the homeostasis of the human body. The influence of diet on the development of asthma is primarily sought in prenatal life. A woman’s nutritional status can therefore affect the immune response of the fetus. The above behaviors consequently deactivate the immunomodulatory processes of the microbiota state according to the microbiota programming of the organism in the first three years of a child’s life. The changes provide a basis for activation of homeostatic disturbances between lymphocytes of the Th1/Th2 type and adaptation of the dominant phenotypic picture in the form of Th2 lymphocytes during the intrauterine life of the fetus [5,10,13,14]. The concept of the allergic march phenomenon (atopic march) highlights the immunological relations between allergic disease entities: asthma and allergy, and atopic dermatitis, providing a basis for an innovative therapeutic model of immunodependent diseases in the prenatal and postnatal period [15,16]. Allergic rhinitis and asthma share common mechanisms of allergic inflammation. The action of environmental factors negatively affects the cells of the airway epithelium, which is not only a barrier against external environmental influences but also participates in the immune response. The epithelium initiates responses to inhaled substances, while epithelium-derived cytokines are responsible for the recruitment and activation of immune cells in the airways. Studies indicate a link between changes in epithelial structure and function and asthma. Half of those diagnosed with asthma had an active type 2 immune response in the airways, resulting in increased mucus production by the epithelium and airway obstruction [17]. However, allergic diseases do not only affect children with asthma.

The purpose of this study was to compare the prevalence of respiratory symptoms and allergies in a group of children with and without asthma and to evaluate the association between exposure to environmental factors and the prevalence of bronchial asthma in a pediatric population.

The following research questions were posed to address the purpose of the study:Is there a difference between the frequency of respiratory symptoms and the diagnosis of asthma?Are allergies and allergic diseases more common in children with bronchial asthma?Does the prevalence of asthma depend on exposure to environmental risk factors such as tobacco smoke, animal allergens, and the presence of mold or moisture in children’s homes?Does a diagnosis of asthma affect a child’s health assessment?

## 2. Materials and Methods

### 2.1. Study Area

A cross-sectional study was conducted on a group of 995 children attending elementary schools in the Silesian province between 2018 and 2019. The largest percentage of respondents included respondents living in cities such as Rybnik (*n* = 397; 40.2%), Racibórz (*n* = 275; 27.8%), Tarnowskie Góry (*n* = 143; 14.5%), Gliwice (*n* = 73; 7.4%), and Żywiec (*n* = 58; 5.9%). The remaining students (4.2%) were residents of Czechowice-Dziedzice, Krupski Mlyn, Zawiercie and Cieszyn.

### 2.2. Characteristics of the Study Group

The study included 995 subjects, including 537 girls (54.0%) and 458 boys (46.0%) aged 7–16 years. The mean age of the students studied was 10.9 ± 2.2 years. The vast majority of respondents were urban residents 966 (*n* = 962; 97.3%); rural residents accounted for 2.7% of respondents. Bronchial asthma affected 88 subjects (8.8%).

### 2.3. Eligibility Criteria

A group sampling method was used to select the sample: localities from particular districts of the Silesian voivodeship were randomly chosen, and then school principals from selected localities were invited to participate in the study.

The basic inclusion criterion was the patient’s written consent, made through participation in the questionnaire. Participation in the study was anonymous and entirely voluntary. The study complies with the provisions of the Helsinki Declaration. The project of the study in light of the Act of 5 December 1996 on the professions of doctor and dentist (*Journal of Laws of 2011, No. 277, item 1634, as amended*) is not a medical experiment and does not require the approval of the Bioethics Committee of the Medical University of Silesia in Katowice. In addition, the data collected were based on an anonymous questionnaire, to which parents gave voluntary consent. Surveys were distributed directly to parents of school-aged children, so the absence of a completed survey meant that they did not agree to participate in the study.

Bronchial asthma was defined based on a positive response to the question, “*has a physician ever diagnosed asthma in a child*”. Bronchitis was defined based on a positive response to the question, “*has a physician ever diagnosed spastic or obstructive bronchitis in a child?*”. Self-assessment of health status was assessed by parents’ responses to the questionnaire: “*How do you assess the health of your child*” with a choice of “*very good*”, “*rather good*”, and “*average*”. In the analysis of environmental risk factors, the following were considered: the presence of mold and moisture in the child’s home, contact with pets, and exposure to tobacco smoke. In addition, the prevalence of selected allergies and allergic diseases such as food and drug allergies, pet dander, pollen, house dust allergies, conjunctivitis, hay fever, and atopic dermatitis were determined.

### 2.4. Research Tool

The research tool was an anonymous questionnaire developed based on the form used in the International Study of Asthma and Allergies in Childhood (ISAAC), with a response rate of 74.3% taking into account the Χ parameter (kappa). We do not ask the parents about any other health outcomes or diseases separate from respiratory diseases. Parents only made self-assessments of their child’s health based on the categories: very good, rather good, and average. The fact that an allergic skin test was performed was assessed by a positive answer to the question, “Has your child ever had an allergic skin test?” The presence of mold and moisture in children’s apartments was also assessed based on parents’ indications in the questionnaire. The questionnaire was completed by the parents of the study children. All data were coded with appropriate symbols preventing the identification of patients by the Act of 29 August 1997, on the Protection of Personal Data (Journal of Laws 1997 No. 133 item 883).

### 2.5. Statistical Analyses

Statistical calculations were performed using the STATISTICA 13.0 program, Stat Soft Poland. Measurable data were characterized using point series, and for non-measurable data, count tables and multivariate tables were used. To assess the relationship between qualitative variables, the chi-square test was used.

The influence of allergy (sensitization to pet allergens, pollen allergy, food, and drug allergies), allergic disease (hay fever, allergic conjunctivitis, atopic dermatitis), and environmental factors on the occurrence of asthma was verified by univariate logistic regression. Unadjusted models (crude odds ratios) with one independent variable of the presence of mold or dampness in the dwelling, exposure to tobacco smoke, and the presence of pets were used. Statistical significance was determined at *p* < 0.05.

## 3. Results

Bronchial asthma was significantly more common in the boys’ group than in the girls’ group (*p* = 0.001). (Table 1). It was also observed that parents’ assessment of their children’s health varied according to the diagnosis of bronchial asthma: when the disease was present, children’s health was most often assessed as rather good or average (*p* < 0.001). We observed a low correlation between subjective assessment of health in comparison with the presence of asthma in children. Detailed analyses are presented in Table 1.

Wheezy ever (*p* = 0.003), dyspnea ever (*p* < 0.001), and bronchitis ever (*p* < 0.001) were significantly more common in the boys’ group than in the girls’ group (Table 2).

We also compared the frequency of respiratory symptoms and bronchitis in children with and without bronchial asthma (Table 3). The most frequently reported respiratory symptoms in all children were dry cough outside of colds in the last 12 months (22.4%) and wheezing ever (20.7%). All analyzed symptoms as well as bronchitis were statistically significantly more frequent in children with ever-diagnosed bronchial asthma than in children without this disease.

Allergic skin tests ever had 435 (43.7%) children performed. Parents of the examined children most frequently declared the occurrence of allergy to pollen (*n* = 186; 19.1%), house dust (*n* = 127; 13.1%), hay fever (*n* = 188; 19.3%), and atopic dermatitis (*n* = 177; 18.1%). All diagnosed allergic reactions and diseases were statistically significantly more common in children diagnosed with bronchial asthma (Table 4).

Analyzing external factors that may contribute to asthma, we observed a similar prevalence of the disease in the group of children exposed (8.9%) and unexposed (8.8%) to tobacco smoke and in the group of children with (9.0%) and without pets (8.7%). However, children whose homes had traces of mold or moisture were more likely to have asthma (11.3%) than children without this exposure (8.2%). However, the differences were not statistically significant (*p* = 0.2). Detailed results of the analyses are presented in Table 5.

A regression analysis (Table 6) confirmed that house dust allergy (OR = 2.6; 95%CI: 1.3–5.1), hay fever (OR = 2.3; 95%CI: 1.2–4.6), and atopic dermatitis (OR = 2.6; 95%CI: 1.6–4.1) increased the risk of asthma.

## 4. Discussion

The results of epidemiological studies have shown that in Poland, asthma affects approximately 12% of the population, or more than 4 million people. The results of the Phase I GAN (Global Asthma Network) study conducted between 2015 and 2020 at multiple centers worldwide, based on the methodology of the ISAAC study, showed that asthma affected 10.5% of children aged 13–14 years and 7.6% of children aged 6–7 years [18]. A similar prevalence of asthma was found in our study: 8.8% of all children were affected. Bronchial asthma is one of the most common chronic diseases of childhood. It is heterogeneous, and its clinical picture, as well as the diagnostic and therapeutic methods used, are different in children and adults. Difficulties in diagnosis and selection of effective and, most importantly, safe medications pose great challenges in the management of this disease [19]. It is estimated that approximately 25–35% of the population has a congenital predisposition to produce excessive amounts of IgE, but only a small percentage of them, only a few percent of the population, develop asthma [20]. Unfortunately, in Poland, asthma is diagnosed too rarely and too late, which poses a great threat and the risk of implementation of inappropriate therapies and the development of many pathological changes in the respiratory system, and, in consequence, permanent damage to the respiratory system [21].

It is also worth mentioning that in the Silesian Province, the incidence of asthma may also be related to worse air quality. It has been proved that there is a connection between the deterioration of air quality typical of winter smog and an increase in the number of registered asthma exacerbations and other respiratory diseases [22,23]. Furthermore, research indicates that even when living in an environment with relatively low levels of air pollution, there is a risk of increased respiratory disease [24]. This has been linked to long-term exposure to air pollution, especially fine particulate matter, nitrogen dioxide, and ozone. These observations may help determine the environmental risk needed to maintain effective efforts to improve environmental air quality. This is consistent with the National Health Program goals related to environmental pollution [24].

The results of our study indicate that parents of children with diagnosed bronchial asthma more often described their health as rather good or average. However, in the case of children without bronchial asthma, parents statistically significantly more often assessed their health as very good. Perhaps this situation results from the simultaneous occurrence of other allergic diseases, which in turn influences the general assessment of the children’s health. This is confirmed by the analyses showing a significantly higher prevalence of allergies and allergic diseases in the group of children with diagnosed bronchial asthma. Respiratory symptoms were also significantly more common in children with asthma than in those without. One in four children with asthma had a chronic cough (lasting at least 3 months), attacks of dyspnea in the last 12 months, wheezing during physical activity, and nearly 40% woke up with wheezing. Thus, it is possible that the combination of these symptoms and diseases influenced the overall health assessment of children diagnosed with bronchial asthma. However, interpretation of this outcome in a cross-sectional study must be cautious due to the limited causal relationship. The general health question may relate not only to respiratory diseases but to many other health outcomes.

In addition, it was observed that parents of children without bronchial asthma also frequently report some respiratory symptoms. As many as 19.9% of children without diagnosed asthma had a dry cough at night in the past 12 months, and 14.5% of children had episodes of wheezing. Symptoms such as chronic cough, dyspnea, and wheezing may be predictors of bronchial asthma, although individual symptoms are not yet indicative of the disease. Previous epidemiological studies have emphasized that the risk of asthma is higher in children who present with any respiratory symptoms, but wheezing may be typical of many diseases with different phenotypes [25]. It is, therefore, important to correctly identify the wheeze phenotype in children at high risk for bronchial asthma.

Due to its nature, the study cannot answer the question of whether the presence of respiratory symptoms in children without diagnosed asthma may indicate a problem of underdiagnosis of this disease. In the Mastalerz-Migasz study, it was indicated that 15% of healthy subjects had a post-exertional cough and nocturnal cough, and 22% of children had upper respiratory tract infections for more than 10 days [26]. Coughing and wheezing are the most common symptoms of this disease. Most attention should be paid to those symptoms occurring at night or in the morning provoked by exercise, laughter, cold air, or tobacco smoke. Cough may often be the only first symptom of mild bronchial asthma. This is clinically important because undiagnosed or poorly treated asthma can have serious consequences [27].

Exposure to extrinsic factors is necessary for the development of full-blown disease. Children are exposed to constant contact with various sensitizing factors that aggravate the course of the disease and provoke the onset of possible symptoms. Asthma attacks can be triggered by infections, dust, psychological and physical stress, or excessive exercise [27]. In our study, environmental risk factors, such as the presence of pets and exposure to tobacco smoke, were not observed to affect the prevalence of bronchial asthma. However, results from other observational studies indicate an indisputable effect of ETS on the risk of developing this disease, with the strongest effect found for in-utero exposure in a maternal smoking situation [28,29]. Exposure to nicotine and ETS affects the cellular and humoral immune response, following increased production of pro-inflammatory cytokines and decreased production of anti-inflammatory cytokines [30]. However, in the case of our study, we cannot exclude the possibility that with the diagnosis of asthma, some parents decided to give up their smoking habit. Perhaps the observed lack of differences in the prevalence of asthma depending on ETS is a consequence in this case. On the other hand, the timeline for smoking in the home and the children’s outcome has not been studied to make this association. A similar observation may apply to the presence of pets, although in this case, the results of observational studies are inconclusive. A cross-sectional study among school-aged children in Hungary found that the prevalence of asthma depended on the type of pet present [31]. The presence of a dog increased the risk of asthma, while a protective effect was observed for rodents. In contrast, owning cats or birds showed no such association. In our study protocol, no question was asked about the type of pets owned, so similar analyses are not possible.

In a study of children from the city of Sosnowiec, Silesia, it was found that there was a statistically significant effect of housing conditions such as dampness in the apartment on the incidence of wheezing and dyspnea attacks in the last 12 months [32]. This observation is also consistent with the results of a previous study conducted in Silesia [33]. The results of our study showed that asthma was more common in the group of children living in houses with visible traces of mold or dampness, but the differences between the groups were not statistically significant. However, this observation should not be underestimated, given the results of other studies confirming the association between mold exposure and the occurrence of bronchial asthma. Moreover, it should be remembered that from the point of view of clinical practice, not only statistical significance but also clinical relevance should be considered.

The findings of our study indicate that house dust allergy, hay fever, and atopic dermatitis increase the risk of asthma. Atopic dermatitis is often one of the first stages of the allergic march that can lead to the development of asthma [34]. Moreover, atopic dermatitis is associated with food allergy and allergic rhinitis. Studies confirm a higher prevalence of asthma in patients with atopic dermatitis [34].

Environmental factors can provoke the development or exacerbation of the disease. However, it should be emphasized that asthma is a highly heterogeneous disease, with a range of endotypes that are associated with chronic airway inflammation. The mechanisms leading to the development of allergic inflammation and asthma are still incompletely elucidated. Attempts to explain this mechanism are explained by the role of innate immune functions and the breakdown of local immune homeostasis. Airway epithelial cells (AECs) and antigen-presenting cells (APCs) produce factors that can enhance or suppress acute inflammatory responses. The formation of pathophysiological features of asthma can occur through the release of pro-inflammatory cytokines such as interleukin (IL) 1β and IL-6, while epithelium-derived IL-33 promotes Th2 differentiation. On the other hand, anti-inflammatory factors are indicated to contribute to the maintenance of homeostasis, including IL-10 and IL-37. A thorough understanding of these mechanisms may be a key factor in the therapy, control, and prevention of bronchial asthma [35].

To answer the question posed in the title, childhood asthma is often associated with other health disorders. It is not only a higher prevalence of respiratory symptoms but also allergic diseases that can determine the quality of life and functioning of children.

## 5. Strengths and Limitations

The strength of the presented study is the large study group. The selection of such a large group significantly minimized the risk of selection bias. Moreover, the questionnaire study was conducted traditionally through a direct interview with the parents of children. Such a procedure allowed us to avoid the common phenomenon of “bot/fake responders”, which is characterized by questionnaires made available based on online forms. A standardized questionnaire was used in the study, which also had a significant impact on the quality of the presented data and the possibility of comparing them with studies conducted based on the ISAAC survey methodology.

This study was conducted before the COVID-19 pandemic, so it created less confusion about other diseases and outcomes that could be confused with asthma symptoms and outcomes. At the same time, it prevented the evaluation of possible other exposures due to social isolation situations. For example, studies have shown that cigarette smoking and smoke exposure may increase the expression of ACE2 receptors in the lower airways. According to a statement by the Italian Society of Childhood Allergy, increased ACE2 expression associated with smoking means increased susceptibility to COVID-19 infection [36]. Current studies support a favorable outlook for children with asthma infected with SARS-CoV-2 [37]. How innate and acquired immunity responds to the virus, the reduced number of virus receptors, and steroid regulation and treatment may play a role in the way SARS-CoV-2 infects children with asthma. Nevertheless, it is not clear that this is the full picture. More knowledge is needed on whether the reduced admissions of asthmatic children with acquired COVID-19 are solely due to the mentioned characteristics or whether there are other parameters [38] and in what role environmental factors, especially environmental pollution, will play. Therefore, further research by the authors will be devoted to this thread.

In future studies, it is planned to undertake an analysis of the above material in comparison with laboratory material. To answer the question: Are there differences in highly determined markers, such as FeNO or IL5, eosinophil counts in blood, or sputum? In the present study, this was not decided, as the use of a standardized tool was relied upon. An epidemiological cross-sectional study was performed using the questionnaire used in the International Study of Asthma and Allergies in Childhood (ISAAC). It is a standardized tool, validated and adapted to the cultural and social context of each country, which allows for a reliable estimation of the prevalence of respiratory diseases and symptoms, especially of childhood bronchial asthma.

The main limitation of the present study, however, was due to the survey model adopted. A cross-sectional study, although it allows for rapid and reasonably accurate data on the health profile of a population, has limited reliability in the existence of a cause-and-effect relationship. Thus, the correlations observed in this study should not be read as a cause-and-effect relationship, as the influence of other factors on the prevalence of asthma, respiratory symptoms, or allergic diseases analyzed cannot be excluded. Nevertheless, this observation does not discredit the present study but only requires proper analysis, interpretation of the data, and drawing of conclusions.

## 6. Conclusions

Asthma prevalence was 8.8% in the pediatric population, with a significantly higher rate of diagnosis in the boy group than in the girl group. The parent’s subjective assessment of the child’s condition varied significantly by asthma diagnosis. Respiratory symptoms, as well as allergies and allergic eye inflammation and AD, were significantly more common in children diagnosed with bronchial asthma. Exposure to environmental factors in the form of tobacco smoke, animal allergens, and the presence of mold and moisture in the home did not significantly affect the prevalence of bronchial asthma.

## Figures and Tables

**Table 1 ijerph-19-11180-t001:** Asthma diagnosis versus gender and subjective health assessment of children.

Asthma		Yes *n* (%)	Not *n* (%)	Total *n* (%)	*p*-Value *
Gender	Girls	33 (6.2%)	504 (93.8%)	537 (54.0%)	0.001
	Boys	55 (12.0%)	403 (88.0%)	458 (46.0%)	
Total *n* (%)		88 (8.8%)	907 (97.2%)	995 (100.0%)	
Subjective health assessment	Very good	31 (35.2%)	532 (58.8%)	563 (56.7%)	<0.001
	Rather good	38 (43.2%)	344 (38.0%)	382 (38.5%)	
	Average	19 (21.6%)	28 (3.1%)	47 (4.8%)	

* *p*-value for chi-square test.

**Table 2 ijerph-19-11180-t002:** Presence of respiratory symptoms or bronchitis versus gender.

Presence of Respiratory Symptoms or Bronchitis	Gender		Total *n* (%)	*p*-Value *
	Girls *n* (%)	Boys *n* (%)		
Cough for at least 3 months (no colds) in the last year	48 (9.0%)	49 (10.7%)	97 (9.8%)	0.3
Dry cough attacks at night in the last 12 months (without colds)	120 (22.3%)	104 (22.7%)	224 (22.5%)	0.9
Wheezy ever	92 (17.1%)	114 (24.8%)	206 (20.7%)	0.003
Wheey in the last 12 months	44 (8.2%)	47 (10.2%)	91 (9.1%)	0.3
Awakenings at night caused by wheezing attacks in the last 12 months	26 (4.9%)	31 (6.8%)	57 (5.7%)	0.2
Attacks of wheezing during physical activity in the last 12 months	28 (5.2%)	24 (5.3%)	52 (5.2%)	0.9
Dyspnea ever	40 (7.4%)	66 (14.4%)	106 (10.7%)	<0.001
Dyspnea in the last 12 months	18 (3.3%)	20 (4.4%)	38 (3.8%)	0.4
Bronchitis ever	66 (12.4%)	96 (21.0%)	162 (16.3%)	<0.001

* *p*-value for chi-square test.

**Table 3 ijerph-19-11180-t003:** Respiratory symptoms and diagnosis of asthma.

Presence of Respiratory Symptoms or Bronchitis	Asthma		Total *n* (%)	*p*-Value *
	Yes *n* (%)	No *n* (%)		
Cough for at least 3 months (no colds) in the last year	22 (25.0%)	74 (8.2%)	96 (9.7%)	*p* < 0.001
Dry cough attacks at night in the last 12 months (without colds)	46 (52.3%)	177 (19.5%)	223 (22.4%)	*p* < 0.001
Wheezy ever	74 (84.1%)	132 (14.5%)	206 (20.7%)	*p* < 0.001
Wheey in the last 12 months	43 (48.9%)	48 (5.3%)	91 (9.1%)	*p* < 0.001
Awakenings at night caused by wheezing attacks in the last 12 months	34 (39.1%)	23 (2.6%)	57 (5.8%)	*p* < 0.001
Attacks of wheezing during physical activity in the last 12 months	21 (24.1%)	31 (3.4%)	52 (5.3%)	*p* < 0.001
Dyspnea ever	53 (60.9%)	52 (5.7%)	105 (10.6%)	*p* < 0.001
Dyspnea in the last 12 months	21 (24.1%)	17 (1.9%)	38 (3.8%)	*p* < 0.001
Bronchitis ever	73 (82.9%)	88 (9.8%)	161 (16.3%)	*p* < 0.001

* *p*-value for chi-square test.

**Table 4 ijerph-19-11180-t004:** Allergies and allergic reactions and the diagnosis of asthma.

Current Allergy or Allergic Disease	Asthma		Total *n* (%)	*p*-Value *
	Yes *n* (%)	No *n* (%)		
House dust allergy	33 (39.3%)	94 (10.6%)	127 (13.1%)	*p* < 0.001
Sensitization of pet allergens	25 (30.9%)	73 (8.3%)	98 (10.2%)	*p* < 0.001
Pollen allergy	42 (47.7%)	144 (16.3%)	186 (19.1%)	*p* < 0.001
Food allergies	19 (22.9%)	944 (10.7%)	113 (11.7%)	*p* < 0.001
Drug allergies	9 (10.8%)	38 (4.3%)	47 (4.9%) **	0.02
Hay fever	40 (47.7%)	148 (16.6%)	188 (19.3%)	*p* < 0.001
Allergic conjunctivitis	28 (31.8%)	94 (10.6%)	122 (12.5%)	*p* < 0.001
Atopic dermatitis	30 (34.1%)	147 (16.5%)	177 (18.1%)	*p* < 0.001

* *p*-value for chi-square test; ** *p*-value for chi-square NW test.

**Table 5 ijerph-19-11180-t005:** Environmental factors and prevalence of asthma.

Environmental Factors		Asthma		Total *n* (%)	*p*-Value *
		Yes *n* (%)	No *n* (%)		
Presence of traces of moisture and mold in the apartment	Yes	23 (11.3%)	181 (88.3%)	204 (20.5%)	0.2
	No	65 (8.2%)	726 (91.8%)	791 (79.5%)	
Environmental exposure to tobacco smoke	Yes	48 (8.9%)	491 (91.1%)	539 (54.2%)	0.9
	No	40 (8.8%)	416 (91.2%)	456 (45.8%)	
Presence of pets	Yes	54 (9.0%)	548 (91.0%)	602 (60.7%)	0.8
	No	34 (8.7%)	355 (91.3%)	389 (39.3%)	

* *p*-value for chi-square test.

**Table 6 ijerph-19-11180-t006:** Crude odds ratios (OR) and 95% confidence intervals related to determinants of observed respiratory symptoms.

Determinants of Asthma	Regression Coefficient	OR (95%CI)	*p*-Value of the Regression Coefficient
House dust allergy (Yes */No)	0.5	2.6 (1.3–5.1)	0.003
Sensitization to pet allergens (Yes */No)	0.2	1.6 (0.8–3.3)	0.2
Pollen allergy (Yes */No)	0.02	1.0 (0.5–2.2)	0.8
Food allergies (Yes */No)	0.1	1.3 (0.7–2.5)	0.4
Drug allergies (Yes */No)	0.4	2.3 (1.0–5.4)	0.05
Hay fever (Yes */No)	0.4	2.3 (1.2–4.6)	0.01
Allergic conjunctivitis (Yes */No)	0.3	1.8 (0.9–3.2)	0.07
Atopic dermatitis (Yes */No)	0.5	2.6 (1.6–4.1)	<0.001
Presence of traces of moisture and mold in the apartment (Yes */No)	0.2	1.4 (0.8–2.3)	0.2
Environmental exposure to tobacco smoke (Yes */No)	0.5	1.0 (0.7–1.6)	0.9
Presence of pets (Yes */No)	0.02	1.0 (0.7–1.7)	0.8

* Reference Group.

## Data Availability

Not applicable.

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
