# Peer review of "Respiratory Symptoms, Allergies, and Environmental Exposures in Children with and without Asthma"

_ijerph, 2022, doi:10.3390/ijerph191811180_

Round 1

Reviewer 1 Report (New Reviewer)

Dear Editors,

the authors improved the manuscript on several parts. However, the reviewer feels that some more information should be included in the introduction and discussion section. The authors should introduce not only the T2 relation in more detail but also the role of airway epithelial cells. Therefore, the authors should mention IL4, which was shown to orchestrate the epithelial polarization. The role of these asthma-associated factors was also shown in a publication on a biomatrix of upper and lower airways. Furthermore, they should also describe the active role of the airway epithelium as it was shown for asthma patients and the suppressive Secretoglobin1A1 in cells of the lower airways as well as IL37 regulating allergic inflammation by counter-balancing IL1 and IL33.

Author Response

Dear Reviewer

Thank you for your review and valuable suggestions. We tried to include all of them, as all mechanisms that can both provoke and weaken the symptoms of the disease are relevant in asthma. We hope that the completed information will be sufficient. We have not described the operation of these mechanisms in detail, as this would not fully correspond to the purpose of this study. However, in order to fully understand the complexity of a disease like asthma and its determinants, we have supplemented the paper with the following passages:

Translated with www.DeepL.com/Translator (free version)Allergic rhinitis and asthma share common mechanisms of allergic inflammation. The action of environmental factors negatively affects the cells of the airway epithelium, which is not only a barrier against external environmental influences, but also participates in the immune response. The epithelium initiates responses to inhaled substances, while epithelium-derived cytokines are responsible for the recruitment and activation of immune cells in the airways. Studies indicate a link between changes in epithelial structure and function and asthma. Half of those diagnosed with asthma had an active type 2 immune response in the airways, resulting in increased mucus production by the epithelium and airway obstruction [18].

Environmental factors can provoke the development or exacerbation of the disease. However, it should be emphasized that asthma is a highly heterogeneous disease, with a range of endotypes that are associated with chronic airway inflammation. The mechanisms leading to the development of allergic inflammation and asthma are still incompletely elucidated. Attempts to explain this mechanism are explained by the role of innate immune functions and the breakdown of local immunohomeostasis. Airway epithelial cells (AECs) and antigen-presenting cells (APCs) produce factors that can enhance or suppress acute inflammatory responses. The formation of pathophysiological features of asthma can occur through the release of pro-inflammatory cytokines such as interleukin (IL) 1β and IL-6, while epithelium-derived IL-33 promotes Th2 differentiation. On the other hand, anti-inflammatory factors are indicated to contribute to the maintenance of homeostasis, including IL-10 and IL-37. A thorough understanding of these mechanisms may be a key factor for the therapy, control and prevention of bronchial asthma [36].

We sincerely hope that the completed information will allow our publication to be considered valuable and accepted for publication in IJERPH.
With best regards, Authors

Reviewer 2 Report (New Reviewer)

The study presents many limitations: The use os a questionnaire limits many of the conclusions such as those related to exposure to allergens without any objective measures. 

There is a need for a full english revision (grammar, spelling)

The data related to subject health assessment are limited. I would suggest to remove them.

The discussion related to limitations of the use of questionaaire must be enriched

Author Response

Dear Reviewer

Thank you for your review and valuable suggestions.

The study presents many limitations: The use os a questionnaire limits many of the conclusions such as those related to exposure to allergens without any objective measures. 

Response: A cross-sectional study is a significant limitation in assessing causality. This was taken into account in planning, analyzing and drawing conclusions from the study. The observed relationships were not treated as a cause-and-effect relationship. In addition, this study model, despite its limitations, is widely used in scientific research and provides a starting point for further in-depth analytical or clinical studies. However, an important element of a cross-sectional study is a properly constructed questionnaire, which increases the reliability of the data obtained. In the case of our study, we used the standardized tool used in the international ISAAC survey.

There is a need for a full english revision (grammar, spelling)

Response: The entire publication has been revised by a native speaker.

The data related to subject health assessment are limited. I would suggest to remove them.

Response: As suggested by previous reviewers, these data should be included in this publication, so despite their subjectivity, they have not been removed. 

The discussion related to limitations of the use of questionaaire must be enriched

Response: The limitation of the cross-sectional survey model is taken into account:

The main limitation of the present study, however, was due to the survey model adopted. A cross-sectional study, although it allows for rapid and reasonably accurate data on the health profile of a population, has limited reliability on the existence of a cause-and-effect relationship. Thus, the correlations observed in this study should not be read as a cause-and-effect relationship, as the influence of other factors on the prevalence of asthma, respiratory symptoms or allergic diseases analyzed cannot be excluded. Nevertheless, this observation does not discredit the present study, but only requires proper analysis, interpretation of the data and drawing of conclusions.

We sincerely hope that the completed information will allow our publication to be considered valuable and accepted for publication in IJERPH.

With best regards, Authors

This manuscript is a resubmission of an earlier submission. The following is a list of the peer review reports and author responses from that submission.

Round 1

Reviewer 1 Report

The article has been corrected according to my suggestions.